

# Genome-wide characterization of the $Zn(II)_2Cys_6$ zinc cluster-encoding gene family in *Pleurotus ostreatus* and expression analyses of this family during developmental stages and under heat stress

Zhihao Hou[1,2], Qiang Chen[1,2], Mengran Zhao[1,2], Chenyang Huang[1,2] and Xiangli Wu[1,2]

[1] Institute of Agricultural Resources and Regional Planning, Chinese Academy of Agricultural Sciences, Beijing, China
[2] Key Laboratory of Microbial Resources, Ministry of Agriculture and Rural Affairs, Beijing, China

## ABSTRACT

*Pleurotus ostreatus* is one of the most widely cultivated mushrooms in China. The regulatory mechanisms of fruiting body formation and the response to heat stress in *P. ostreatus* are main research focuses. The $Zn(II)_2Cys_6$ family is one of the largest families of transcriptional factors and plays important roles in multiple biological processes in fungi. In this study, we identified 66 zinc cluster proteins in *P. ostreatus* (PoZCPs) through a genome-wide search. The PoZCPs were classified into 15 types according to their zinc cluster domain. Physical and chemical property analyses showed a huge diversity among the PoZCPs. Phylogenetic analysis of PoZCPs classified these proteins into six groups and conserved motif combinations and similar gene structures were observed in each group. The expression profiles of these PoZCP genes during different developmental stages and under heat stress were further investigated by RNA-sequencing (RNA-seq), revealing diverse expression patterns. A total of 13 PoZCPs that may participate in development or the heat stress response were selected for validation of their expression levels through real-time quantitative PCR (RT-qPCR) analysis, and some developmental stage-specific and heat stress-responsive candidates were identified. The findings contribute to our understanding of the roles and regulatory mechanisms of ZCPs in *P. ostreatus*.

# INTRODUCTION

*Pleurotus ostreatus* is a widely cultivated mushroom in China because of its high nutritional value, uncomplicated cultivation techniques, and broad adaptability (*Miles & Chang, 2004*; *Khan & Tania, 2012*). In 2017, the annual yield of *P. ostreatus* in China was 5.46 tons, which was the third largest yield among all edible mushrooms, according to the statistics of the Chinese Edible Fungi Association. Because cultivation of *P. ostreatus* is mainly carried out in traditional greenhouses without environmental control systems, extreme and continuous high temperature always result in "spawn-burning" and easily

Corresponding authors
Chenyang Huang,
huangchenyang@caas.cn
Xiangli Wu, wuxiangli@caas.cn

causes infection by green mold, which ultimately affects quality and yield (*Zhang et al., 2016a*; *Qiu et al., 2017*). The mechanisms of the heat stress response in mushrooms have become a hot research topic, with studies focusing on signaling molecules (*Zhang et al., 2016b*; *Liu et al., 2018*), thermoprotective components (*Wang et al., 2018a*; *Lei et al., 2019*; *Liu et al., 2019*), antioxidant enzymes (*Wang et al., 2017*), and morphological characteristics (*Song et al., 2014*). Furthermore, transcription factors play important roles in heat stress responses. For example, heat shock transcription factors (HSFs) regulate heat tolerance in wheat and *Arabidopsis thaliana* (*Jiang et al., 2018*; *Tian et al., 2019*). Dehydration responsive element binding protein (DREB) improves heat resistance in *A. thaliana* and *Nicotiana tabacum* (*Sakuma et al., 2006*; *Arroyo-Herrera et al., 2016*). Transcription factors such as those of the NAC (*Shahnejat-Bushehri, Mueller-Roeber & Balazadeh, 2012*; *Fang et al., 2015*; *Guo et al., 2015*), MYB (*El-Kereamy et al., 2012*; *Casaretto et al., 2016*), WRKY (*Li et al., 2009*; *Cai et al., 2015*; *He et al., 2016*) and zinc finger (*Kim, Cho & Yoo, 2015*; *Koguchi et al., 2017*; *Agarwal & Khurana, 2018*) families are involved in tolerance to high temperature in many eukaryotes.

Fruiting bodies are arguably the most complex multicellular structures that are produced by fungi, and it is important to gain further insight into fruiting body morphogenesis (*Nowrousian, 2018*). Transcription factors, such as SQUAMOSA promoter-binding protein (SBP), homeodomain transcription factors and NAC, are involved in a variety of developmental processes (*Cardon et al., 1999*; *Ernst et al., 2004*; *Vonk & Ohm, 2018*).

The $Zn(II)_2Cys_6$ protein family, which is also called the zinc cluster protein (ZCP) family, is a large transcription factor family found exclusively in fungi. Members of these families bind two zinc atoms with a DNA-binding domain consisting of six cysteine residues (*Vallee, Coleman & Auld, 1991*; *MacPherson, Larochelle & Turcotte, 2006*). Previous studies have shown that ZCPs act as multifunctional regulators in many biological processes. Indeed, ZCPs have been shown to participate in primary metabolism, including sugar, amino acid, vitamin and uracil metabolism (*Losson & Lacroute, 1981*; *Nishimura et al., 1992*; *Iraqui et al., 1999*), as well as in secondary metabolism, such as ergosterol biosynthesis and melanin biosynthesis (*Crowley et al., 1998*; *Tsuji et al., 2000*). Moreover, ZCPs have been implicated in fungal development and the response to stresses, such as heat shock, oxidative stress and high osmotic stress (*Akache, Wu & Turcotte, 2001*; *Ohm et al., 2011*; *Schumacher et al., 2018*).

The ZCP gene family has been identified in many fungi, and it is well characterized in *Saccharomyces cerevisiae* (*Akache, Wu & Turcotte, 2001*), *Candida albicans* (*Schillig & Morschhäuser, 2013*), *Aspergillus flavus* (*Chang & Ehrlich, 2013*) and *Tolypocladium guangdongense* (*Zhang et al., 2019*). However, similar studies in Agaricales and even basidiomycetes are, to our knowledge, very limited. Genome-wide identification of ZCP genes in different species not only provides useful information regarding the biological functions of this important gene family but also reveals targets for molecular breeding. In this study, ZCP genes in *P. ostreatus* were identified by a genome-wide method, and classification and domain analysis were performed. Furthermore, protein properties, phylogenetic analyses, motif distributions and gene architectures among members of the

 

PoZCP family were assessed. Finally, the expression patterns of PoZCP genes during different developmental stages and under heat stress at different time points were investigated based on previously obtained RNA-seq data. Additionally, ten heat-responsive and six development-related PoZCPs were selected for validation of their expression levels through real-time quantitative PCR (RT-qPCR) analysis. In conclusion, this study offers useful information about the PoZCP gene family, and some potential family members that might be involved in development and heat stress were identified, providing targets for molecular breeding of *P. ostreatus*.

## MATERIALS AND METHODS

### Identification of zinc cluster genes

The zinc cluster domain (PF00172) was downloaded from the Pfam database (http://pfam.xfam.org/) (*El-Gebali et al., 2018*). Then, an HMM search was performed using the hmmsearch tools embedded in HMMER3.21 (*Wheeler & Eddy, 2013*) with default parameters based on the protein sequences of *P. ostreatus* strain CCMSSC0389 (China Center for Mushroom Spawn Standards and Control, Beijing, China) (*Qu et al., 2016*). The zinc cluster domain region of each potential positive sequence was extracted and multialigned using the Clustal Omega online tool (*Madeira et al., 2019*). Sequences without complete six cysteine residues in the zinc cluster domain were regarded as false positive sequences and removed. The gene structures of the remaining sequences were corrected by checking the mapped reads positions in the RNA-seq data using the Integrative Genomics Viewer tool (*Robinson et al., 2011*). After manual removal of false positive sequences and correction, the remaining sequences were submitted to the NCBI Conserved Domain Database (https://www.ncbi.nlm.nih.gov/cdd) and the PFAM database to determine the presence and integrity of the zinc cluster domain. Transcription factor annotation was performed by the hmmsearch tool (E-value cutoff = 0.01) using 52 transcription factor Pfam models (Table S1).

### Gene structure and protein analyses of the zinc cluster family

Gene structures were obtained by exon and intron information from the *P. ostreatus* genome. PoZCPs were classified according to the number of amino acid residues between each two cysteine residues in the typical zinc cluster domain. The isoelectric point (pI) and molecular weight (MW) of PoZCPs were estimated using IPC tools (*Kozlowski, 2016*). The subcellular localization of each PoZCP was predicted by BaCelLo online tools (*Pierleoni et al., 2006*).

### Sequence alignment, motif analysis and phylogenetic tree construction of the zinc cluster family

The domain region of the zinc cluster sequences was characterized by the NCBI–CDD and Pfam results and classified by the amino acid counts of variable regions. Multiple sequence alignments of the protein sequences of PoZCPs were carried out using the Clustal Omega online tool (*Madeira et al., 2019*), and the poorly aligned regions were removed by the alignment utility trimAl (*Capella-Gutiérrez, Silla-Martínez & Gabaldón, 2009*).
The aligned sequences were used to construct a phylogenetic tree based on the maximum likelihood (ML) method by IQ-TREE using the best fitting model and branch support computed with 1000 UltraFast Bootstrap (*Trifinopoulos et al., 2016*). Conserved motifs in the PoZCP family were assessed by the MEME online program (*Bailey et al., 2009*). The gene structure and motif distribution of the PoZCP family were displayed using Tbtools (*Chen et al., 2018*).

### RNA-sequencing and analysis

RNA-sequencing (RNA-seq) was conducted to gain insight into the expression patterns of the zinc cluster genes at several developmental stages and heat stress. Fruiting bodies of the dikaryotic strain CCMSSC0389 were grown, and samples, including mycelia, primordia and fruiting bodies, were collected at different developmental stages as described previously (*Wang et al., 2017*). The mycelia were inoculated on PDA medium and cultured at 28 °C for 6 days and then transferred to 35 °C for 0.5, 1, 2 and 3 h, respectively, for heat stress treatment. Three replicates for each sample were frozen in liquid nitrogen immediately after collection.

Sample collections, RNA extraction and RNA-seq were carried out as previously described (*Wang et al., 2018b*), except that TruSeq Stranded mRNA LTSample Prep Kit (Illumina, San Diego, CA, USA) was used for sequencing library construction and the Illumina HiSeq 2500 platform was used for sequencing. Transcript expression of the zinc cluster genes was analyzed according to the RNA-seq profiles using fragments per kilobase per million mapped reads (FPKM) values. A heatmap was constructed using Tbtools with log and row scales (*Chen et al., 2018*).

### Validation of gene expression by RT-qPCR

To validate the RNA-seq results, we performed an RT-qPCR analysis of 13 genes that showed different expression patterns and large fold changes during various developmental stages or under heat stress. The experimental procedures were described in previous studies (*Wang et al., 2017*, *2018b*). Beta-actin and beta-tubulin were used as reference genes. The primer sequences are listed in Table S2. The relative expression level of each gene was calculated using the $2^{-\Delta\Delta Ct}$ method (*Livak & Schmittgen, 2001*).

## RESULTS

### Identification of ZCP genes in *P. ostreatus*

After manual correction, a total of 66 unique genes with conserved zinc cluster domains were identified as ZCP genes in *P. ostreatus*, accounting for 23.82% of all transcription factors. The 66 genes were named *PoZCP01* to *PoZCP66*, and the coding sequences were submitted to GenBank under accession numbers MN652925–MN652990. Fifty-nine (89.39%) family members contain zinc cluster domains in the N-terminus (front 1/3 part of the protein), three (4.55%) family members have zinc cluster domains in the middle (1/3–2/3 of the protein) of the protein, and four (6.06%) family members possess zinc cluster domains in the C-terminus (last 1/3 part of the protein) (Fig. 1). Fungal-specific transcription factor domains (Fungal_trans, PF04082) are present in 43 (65.15%) family
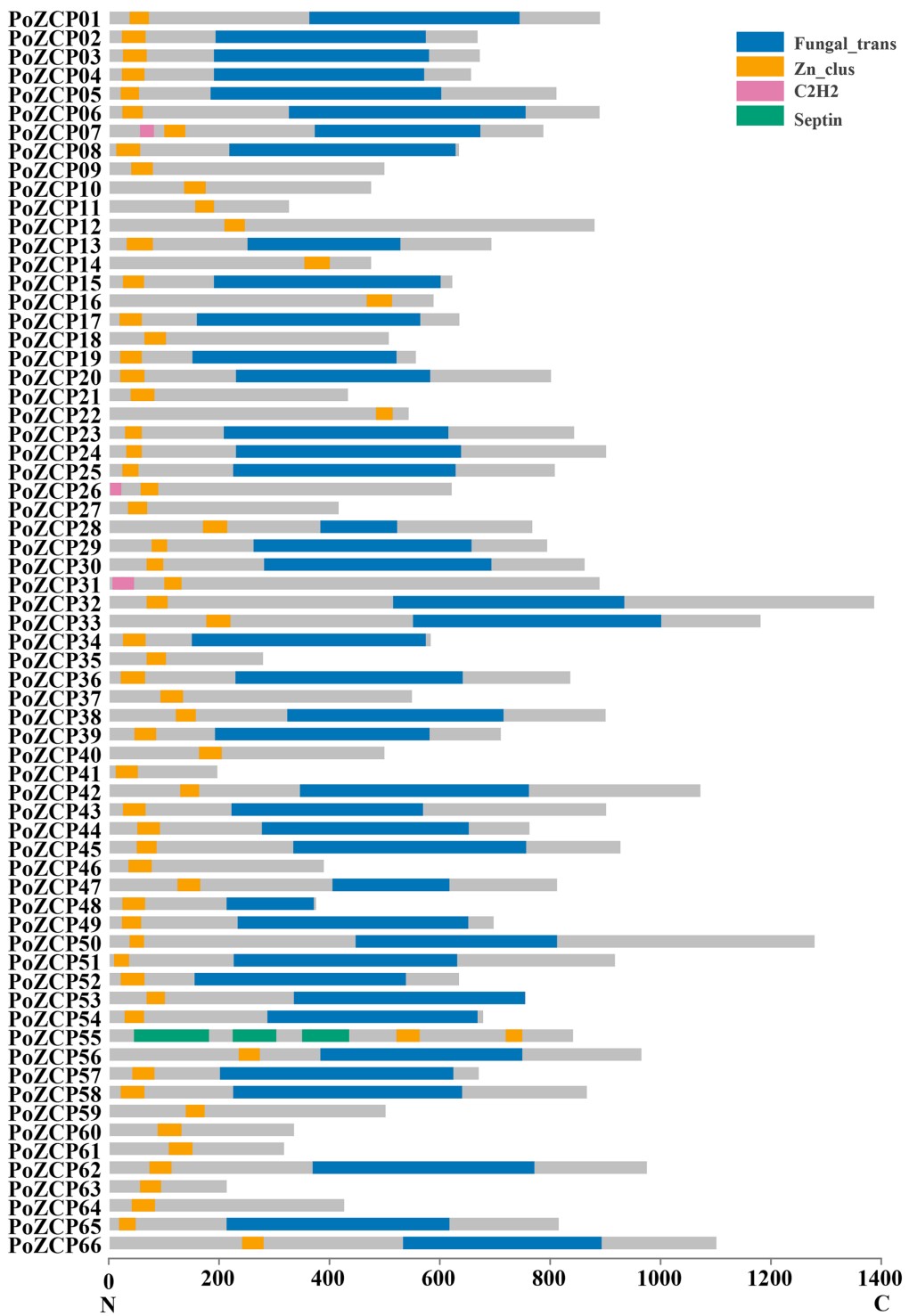

**Figure 1 The distribution of conserved domains in 66 members of the PoZCP family.** The domain organization of the zinc cluster proteins was determined from the Pfam and NCBI conserved domain databases. The *X*-axis represents the amino acid position from the N-terminus to the C-terminus. The gray rectangles represent amino acid regions with no identified domain, and the rectangles in different colors represent different domains, as shown in the key.

members, and C2H2-type zinc transcription factor domains (PF00096) are also present in three family members. One gene, *PoZCP55*, encodes three septin domains (PF00735), which are reported to be involved in cytokinesis and septum formation (*Longtine et al., 1996*), with two zinc cluster domains in the C-terminus.

## Classification and characterization of ZCP genes in *P. ostreatus*

Previous studies in yeast have revealed that in the zinc cluster domain, the numbers of amino acids between C1–C2, C2–C3 and C4–C5 are always 2, 6 and 2, respectively, but that numbers between C3–C4 and C5–C6 vary (*Todd & Andrianopoulos, 1997*). The PoZCP family was classified into 15 subgroups named types A-O based on the different patterns between C3–C4 and C4–C5 (Table 1), 10 of which belong to the typical zinc cluster motif $C–X_2–C–X_6–C–X_{5-12}–C–X_2–C–X_{6-8}–C$ in yeast (*MacPherson, Larochelle & Turcotte, 2006*). However, five other patterns, including $C–X_2–C–X_6–C–X_5–C–X_2–C–X_4–C$ (type A), $C–X_2–C–X_6–C–X_6–C–X_2–C–X_9$ (type G), $C–X_2–C–X_6–C–X_{13}–C–X_2–C–X_6$ (type M), $C–X_2–C–X_6–C–X_{14}–C–X_2–C–X_6$ (type N) and $C–X_2–C–X_6–C–X_{15}–C–X_2–C–X_6$ (type O), were also found in the PoZCP family (Table 1). The motifs of 15 types of zinc cluster domains were analyzed, and the motif logos of the seven major types (more than three members) are shown in Fig. 2. For types A, D, F, I and J, amino acids are conserved at position 5 (arginine) and position 8 (lysine); for types A, B, C and F, amino acids at position 12 (aspartate) and position 16 (proline) are conserved (Fig. 2).

The characteristics, including exon and intron length, protein length, isoelectric point (pI), molecular weight (MW) and predicted subcellular localization by BaCelLo online tools are shown in detail in Table S3. The protein length of putative PoZCP genes ranges from 195 to 1,386 amino acids, with MWs ranging from 21 to 152 kDa and theoretical pIs ranging from 4.86 to 9.45, revealing significant differences among PoZCPs in terms of their physical and chemical properties. According to the predicted subcellular localization, 62.12% of PoZCPs are located in the nucleus, 31.34% in the cytoplasm and 6.06% in secretory pathways (Table S3).

## Phylogenetic, motif and structure analysis of PoZCPs

A phylogenetic tree was constructed using amino acid sequences of the 66 PoZCP family members. The family members were divided into six well-conserved groups (I–VI) with significant bootstrap support according to phylogenetic relationships (Fig. 3A). PoZCP55 forms a separate clade, which imply its self-dependent origin. The PoZCP family members belonging to types A, B, C and F (Table 1) clustered respectively, suggesting genes in each class may have expanded from a single gene. Conversely, PoZCP family members of types D, I and J (Table 1) were found to be distributed across different clades, indicating that these categories may have different ancestors.

To further investigate the diversity of the ZCP in *P. ostreatus*, we analyzed PoZCP motifs using the MEME online server. Ten conserved motifs named motif 1 to motif 10 were identified. Motif 1, which forms the zinc cluster domain, was the most common motif found in all family members. The combinations of different motifs differed among the

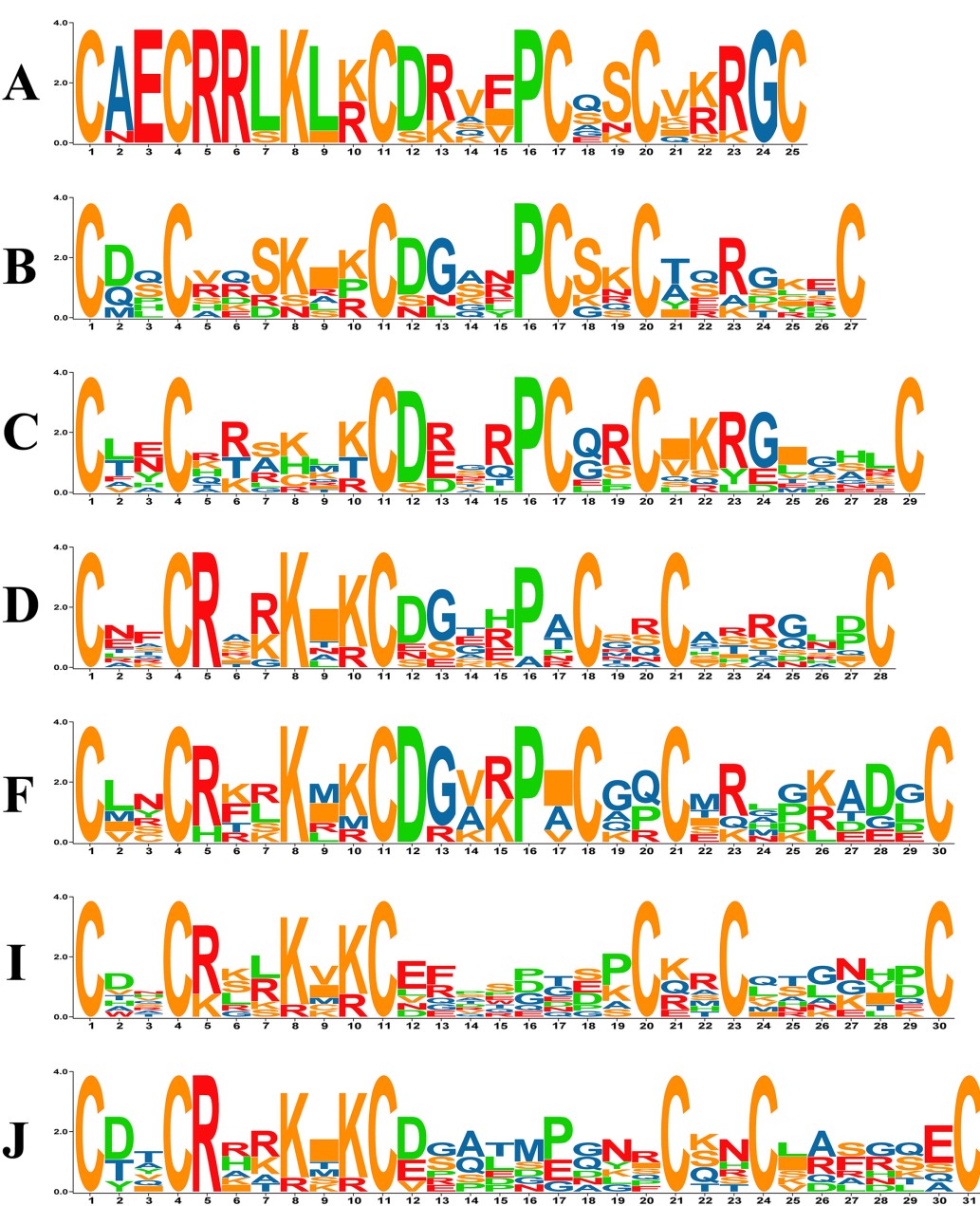

**Figure 2 The motif logos of major types of zinc cluster domains in the PoZCP family.** The capital letters (A–D, F, I, J) on the left side of the logos represent the PoZCP family types shown in Table 1. The X-axis represents the positions beginning with the first cysteine of the zinc cluster domain, and the Y-axis represents the number of bits for each amino acid. The amino acids are shown with one-letter abbreviations.                                 

groups (Fig. 3A). For example, the combination of motif 3, motif 7 and motif 9 occurs in most proteins of group II, the combination of motifs 7–9 appeared in most proteins of group IV, the combination of motif 3, motif 7 and motif 8 existed in all members of group V and motifs 2–6 were found in all members of group VI (Fig. 3B). These results suggest that the function of the PoZCPs of different groups may differ.

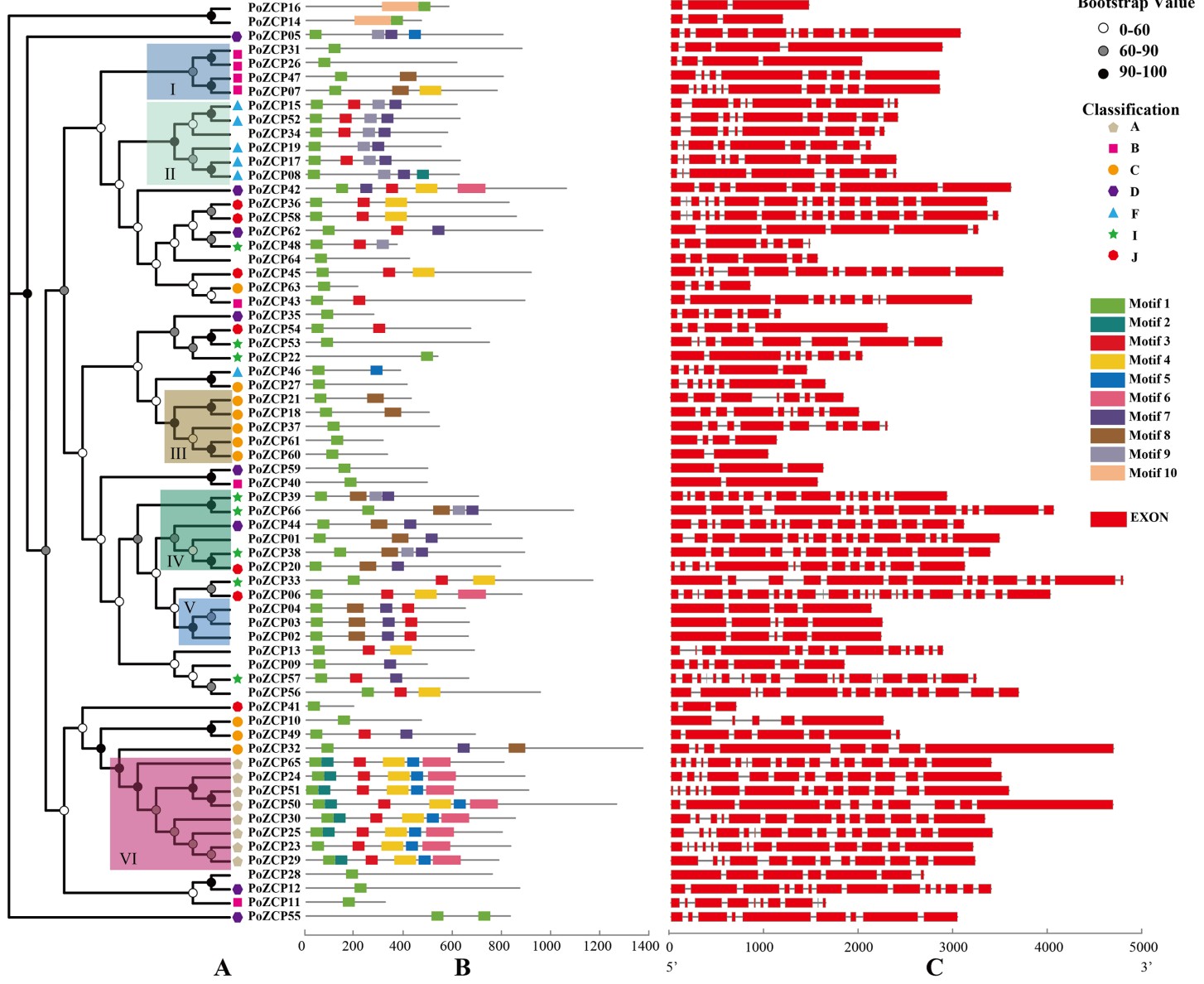

**Figure 3 Phylogenetic tree, motif composition and gene structures of the PoZCP family.** (A) Phylogenetic tree obtained using the maximum likelihood (ML) method by IQ-TREE with best fitting model branch support computed with 1000 UltraFast Bootstrap. Different colors of nodes represent different ranges of bootstrap support values. The classification of different types of PoZCPs is shown with polygons of different shapes and colors. (B) Motif composition of the PoZCP proteins, obtained by the MEME program. The gray line represents the length of the proteins, and the rectangles in different colors represent different motifs from motif 1 to motif 10. (C) Gene structures of PoZCP genes. The exons are represented by red rectangles, and the black lines connecting two exons represent introns.

We deduced the gene structures of individual PoZCP genes to examine the relationship between their evolutionary origins and structural diversity. The results showed that the exon number of PoZCP genes changed over a wide range (from 2 to 23); however, each group in Fig. 3A is relatively conserved in gene structure (Fig. 3C). For example, group II consists of genes of approximately 2,500 bp with similar architecture and 8–10 exons that encode approximately 600 amino acids. Group III is composed of short genes of

**Table 1 Classification of PoZCP family members.**

| Type | Pattern | Counts |
|------|---------|--------|
| A | $C–X_2–C–X_6–C–X_5–C–X_2–C–X_4–C$ | 8 |
| B | $C–X_2–C–X_6–C–X_5–C–X_2–C–X_6–C$ | 7 |
| C | $C–X_2–C–X_6–C–X_5–C–X_2–C–X_8–C$ | 10 |
| D | $C–X_2–C–X_6–C–X_6–C–X_2–C–X_6–C$ | 8 |
| E | $C–X_2–C–X_6–C–X_6–C–X_2–C–X_7–C$ | 1 |
| F | $C–X_2–C–X_6–C–X_6–C–X_2–C–X_8–C$ | 6 |
| G | $C–X_2–C–X_6–C–X_6–C–X_2–C–X_9–C$ | 2 |
| H | $C–X_2–C–X_6–C–X_7–C–X_2–C–X_6–C$ | 1 |
| I | $C–X_2–C–X_6–C–X_8–C–X_2–C–X_6–C$ | 8 |
| J | $C–X_2–C–X_6–C–X_9–C–X_2–C–X_6–C$ | 7 |
| K | $C–X_2–C–X_6–C–X_{10}–C–X_2–C–X_6–C$ | 1 |
| L | $C–X_2–C–X_6–C–X_{11}–C–X_2–C–X_6–C$ | 3 |
| M | $C–X_2–C–X_6–C–X_{13}–C–X_2–C–X_6–C$ | 1 |
| N | $C–X_2–C–X_6–C–X_{14}–C–X_2–C–X_6–C$ | 2 |
| O | $C–X_2–C–X_6–C–X_{15}–C–X_2–C–X_6–C$ | 1 |

**Note:**
The classification of different types of PoZCP family members was based on the variable regions of different zinc cluster domains. The variable regions are shown as underlined characters.

approximately 1,600 bp that encode approximately 450 amino acids. Group IV and group VI are rich in exons, numbering approximately 14, which is significantly greater than the average exon number of 9 for all family members. Most of the genes in group VI contain many short exons of fewer than 50 bp.

## Expression patterns of *PoZCP*s during different developmental stages and under heat stress

To gain insight into the expression profiles of PoZCP genes during different developmental stages and under heat stress at different time points, RNA-seq data obtained from different developmental stages as well as different heat treatment times were used to analyze expression patterns (Tables S4 and S5). *PoZCP14* was not expressed in any expression profile, indicating that it may be a pseudogene.

The expression patterns during different developmental stages were categorized into four clusters according to RNA-seq profiles (Fig. 4A). Genes in cluster I and cluster IV were downregulated and upregulated, respectively, from the mycelial to fruiting body development stages (Fig. 4A), which suggests that these genes may act as negative or positive regulators during development. Genes in cluster II and cluster III were only upregulated in the fruiting body stage and primordia stage, respectively (Fig. 4A), therefore, these genes might be involved in the stage-specific process.

The expression patterns under heat stress at different time points were also categorized into four groups (Fig. 4B). Clusters I and IV were downregulated after heat stress treatment at different time points, which indicates their negative regulatory ability under long-term (2–3 h) and short-term (0.5–1 h) heat stress, respectively (Fig. 4B). Most genes
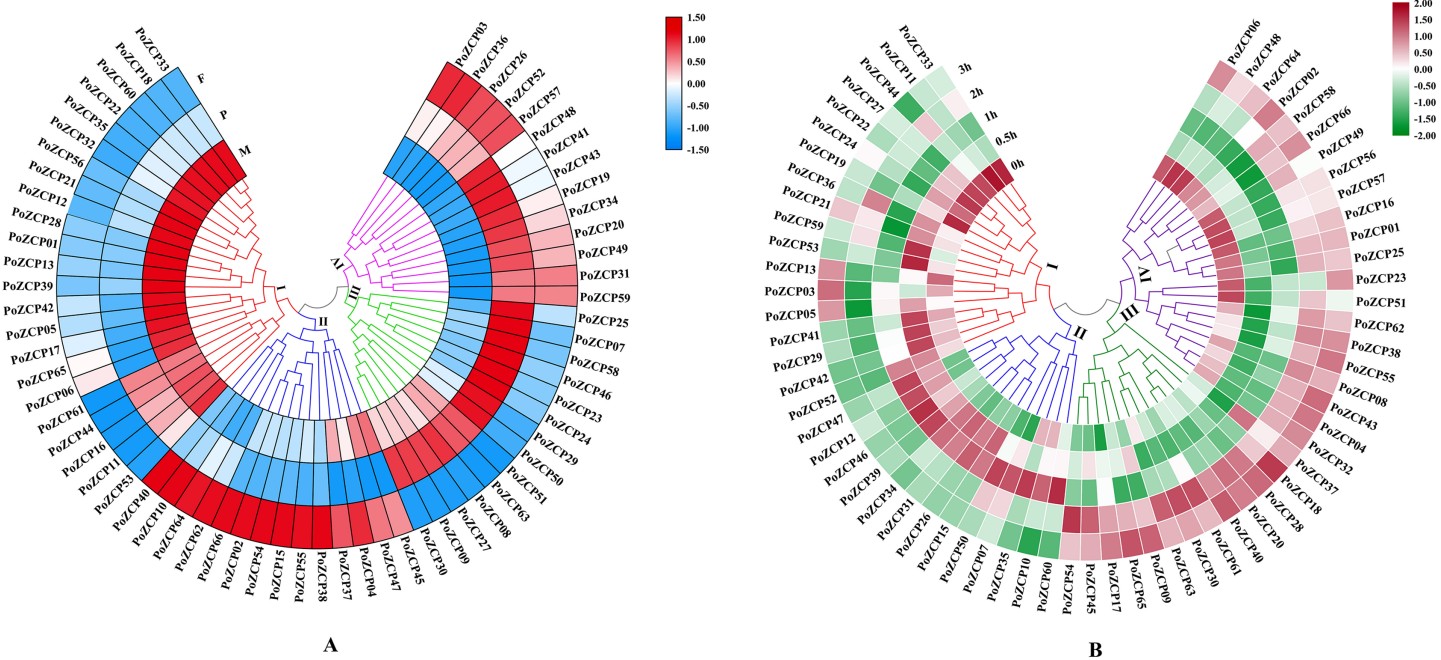

**Figure 4 Expression levels of PoZCPs under different conditions using RNA-seq analysis.** (A) Expression levels of PoZCPs during different developmental stages. M, mycelia (control); P, primordia; F, fruiting bodies. The heatmap was constructed based on RNA-seq analysis of PoZCP genes. The color scale at the top right corner represents high to low expression levels of each gene from red to blue. Different colors of clades represent different expression patterns. (I), Downregulated during developmental stages; (II), upregulated at the fruiting body stage; (III), upregulated at the primordia stage; (IV), upregulated during developmental stages. (B) Expression levels of PoZCPs under heat stress at different time points. The heatmap was constructed based on RNA-seq analysis of PoZCP genes. Mycelia without heat treatment (0 h) were analyzed as a control. The color scale at the top right corner represents high to low expression levels of each gene from red to green. Different colors of clades represent different expression patterns. (I), Downregulated during heat stress; (II), upregulated during 0.5–1 h of heat treatment. (III), upregulated during 2–3 h of heat treatment; (IV), downregulated during 0.5–1 h of heat treatment.

in clusters II and III were upregulated after heat stress treatment at different time points, revealing their positive regulatory ability under short-term (0.5–1 h) and long-term (2–3 h) heat stress, respectively (Fig. 4B).

To verify the gene expression results for the identified PoZCPs, thirteen genes showing distinct variation at various developmental stages or heat treatment in RNA-seq analysis were selected for detection of their expression levels via RT-qPCR. All tested genes showed the same tendency as the transcript profiles (Fig. 5). *PoZCP10, PoZCP26, PoZCP31*, and *PoZCP40* were dramatically upregulated (20–90-fold) in the fruiting body stage, suggesting that they are stage-specific genes. *PoZCP15, PoZCP26*, and *PoZCP31* were markedly upregulated (5–140-fold) under 0.5–1 h heat stress (Fig. 5), which indicates their important roles in the response to short-term heat stress. *PoZCP26* and *PoZCP31* were upregulated during both developmental stages and under heat stress treatment and thus may have multiple functions in development and the heat stress response.

## DISCUSSION

Transcription factors play important roles in growth and development as well as in responses to various abiotic stresses by regulating downstream genes (*Wu & Gallagher, 2012*).

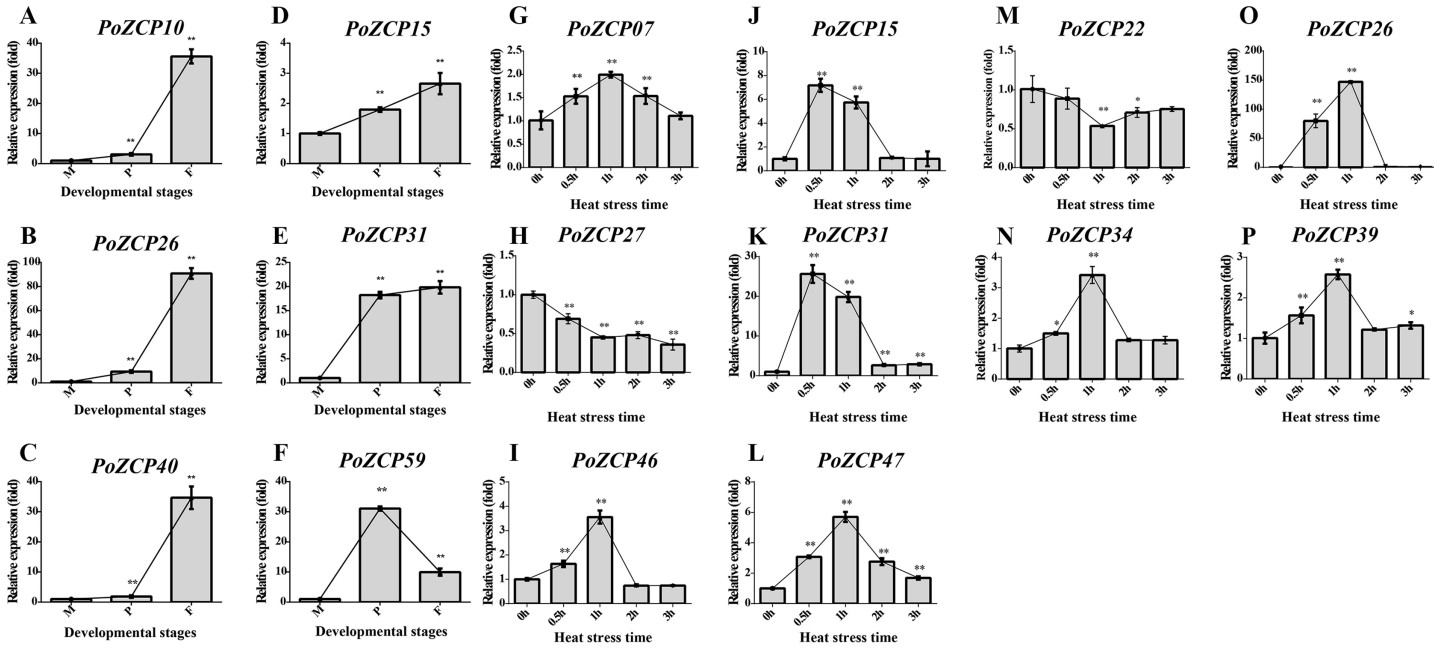

**Figure 5 Expression analysis of 13 PoZCP genes using RT-qPCR.** (A–I) The expression levels of six PoZCP genes during different developmental stages. M, mycelia (control); P, primordia; F, fruiting bodies. (J–P) The expression levels of 10 PoZCP genes under heat stress. Mycelia without heat treatment (0 h) were analyzed as a control. The data are representative of three independent experiments. The error bars represent the standard deviations of three replicates. Statistical analysis was performed using the t test. An asterisk (*) represents $p < 0.05$ and two asterisks (**) represent $p < 0.01$.

As the genomes of an increasing number of organisms have been sequenced, these sequences have supplied valuable support for the discovery of potential transcription factors that may contribute to important agronomic traits (*Li et al., 2014*). The ZCP family, which is the largest transcription factor family in many ascomycetes and basidiomycetes (*Todd et al., 2014*), has been previously found specifically in fungi. In this study, we identified 66 members of the ZCP family in *P. ostreatus*, which accounted for 23.82% of all transcription factors. In a previous study, 55 members of this family were identified in *S. cerevisiae* (*Joshua & Höfken, 2017*), and genome-wide analysis showed that 82 putative ZCPs exist in *C. albican*s (*Schillig & Morschhäuser, 2013*). For the pathogenic fungus *A. flavus* and medicinal fungus *T. guangdongense*, up to 304 and 139 family members, respectively, have been identified (*Chang & Ehrlich, 2013*; *Zhang et al., 2019*). The amount and proportion of PoZCPs is similar to the average for basidiomycetes, at 58 (20.49%), based on 31 basidiomycete genomes, but far below the average from ascomycetes, at 121 (33.43%), based on 77 ascomycete genomes. This finding suggests that this family developed after the split from ascomycetes, and that the underlying molecular mechanisms may be different (*Todd et al., 2014*). Most zinc cluster motifs are located in the N-terminus of the protein. However, four PoZCPs, including PoZCP14, PoZCP16, PoZCP22 and PoZCP55, are C-terminal ZCPs. At least two characterized C-terminal ZCPs also exist in *S. cerevisiae* and *C. albicans* (*Whiteway, Dignard & Thomas, 1992*).

Zinc cluster motifs display a wide spacing pattern of $C–X_2–C–X_6–C–X_{5-12}–C–X_2–C–X_{6-8}–C$, similar to that observed in yeasts. As shown in this study, 15 types of zinc

cluster motifs, including five from different types against yeast, are found in PoZCPs. These findings are similar to those in *Aspergillus* species and *T. guangdongense*, which also exhibits several specific patterns of zinc cluster motifs (*Chang & Ehrlich, 2013*; *Zhang et al., 2019*). In addition to the six cysteine residues that coordinate the zinc ions, some amino acids, including arginine and lysine between C2 and C3 and aspartic acid and proline between C3 and C4, are also conserved in different types of zinc cluster patterns (Fig. 2). These residues play an important role in the formation of functional structures in *S. cerevisiae*. For example, the proline located in the loop between the two substructures confers the flexibility to the loop, and the lysine and arginine between C3 and C4 are involved in homodimerization (*Marmorstein et al., 1992*).

Diversity in physical, chemical and structural characteristics is very common among transcription factors. For example, the CCCH zinc finger family presents a wide variation of gene structures in *Brassica rapa* (*Pi et al., 2018*), and the physicochemical parameters of the ZCP family in *T. guangdongense* are also highly divergent (*Zhang et al., 2019*). The characteristics of PoZCPs, including protein length, pI, MW, exon and intron length and number, show variation over a wide range, suggesting high diversity and multiple functions among ZCP genes. Regarding subcellular localization, the proportions of PoZCPs located in the nucleus, cytoplasm and secretory pathways were predicted to be 62.12%, 31.34% and 6.06%, respectively. This proportion is very similar to the ZCPs in *T. guangdongense* with 62.6% in the nucleus, 23% in the cytoplasm and 5% in the secretory pathway (*Zhang et al., 2019*). The ZCPs localized in the cytoplasm may be transported to the nucleus when they participate in transcriptional regulation (*MacPherson, Larochelle & Turcotte, 2006*). The phylogenetic relationship of the PoZCP family can be divided into six groups based on the phylogenetic tree. Some types of PoZCPs were distributed in the same clade, indicating that they have originated from a single gene. In contrast, some types were distributed among all branches of the tree, which suggests that they may have distinct origins. The motifs in PoZCPs are very abundant, but only Motif 1 correlated with the Pfam domain. The potential functions of the other motifs are worthy of further investigation.

In previous studies, homologs of the zinc cluster transcription factor PRO1 were found to be required for fruiting body development in a number of ascomycete species, such as *A. nidulans* (*Vienken & Fischer, 2006*), *Cryphonectria parasitica* (*Sun, Choi & Nuss, 2009*) and *Sordaria macrospora* (*Steffens et al., 2016*). In basidiomycetes, the transcription factors Fst3 and Fst4 of the ZCP family were shown to inhibit and induce mushroom development, respectively, in *Schizophyllum commune* (*Ohm et al., 2011*). We found that the expression patterns during different developmental stages differed among PoZCP family members. Many developmental stage-specific genes were up- or downregulated at specific developmental stages (Fig. 4A). For example, *PoZCP10*, *PoZCP26* and *PoZCP31* were upregulated (20–90-fold) at the fruiting body stage, and *PoZCP59* was upregulated (30-fold) in the primordia stage compared to the mycelia stage (Fig. 5). The gene function and downstream genes that are regulated during fruiting body formation should be further researched.

Many ZCPs have been presumed to be involved in the response to many environmental factors in different species. For example, Stb5 is a key player in the resistance to oxidative stress, and HAL9 is an important factor involved in salt tolerance in *S. cerevisiae* (*Larochelle et al., 2006*). Moreover, many members of the ZCP family in *Cordyceps militaris* (*Yang et al., 2016*) and *T. guangdongense* (*Zhang et al., 2019*) have an active role in the light response. The ZCP YFL052W regulates the thermal sensitivity of yeast on rich medium (*Akache, Wu & Turcotte, 2001*). The expression profiles of PoZCP genes under heat stress at different time points showed that some family members actively participate in the heat stress response. For example, *PoZCP15, PoZCP26* and *PoZCP31* were dramatically upregulated (5–140-fold) under heat stress for 0.5–1 h, though their roles in the response to heat stress remain to be further investigated.

## CONCLUSIONS

This study constitutes an exhaustive identification, characterization and expression analysis of the zinc cluster type transcription factor in the widely cultivated mushroom *P. ostreatus*. The findings provide useful information about the PoZCP gene family that can be utilized in other basidiomycetes. Some genes were identified as candidates for further study because of their potential functions in fruiting body formation and the heat stress response, which may provide targets for molecular breeding of *P. ostreatus*.

### Funding
This research was funded by the National Natural Science Foundation of China (31601803), the Fundamental Research Funds for Central Non-profit Scientific Institution (No. 1610132020032) and the China Agriculture Research System (CARS20). The funders had no role in study design, data collection and analysis, decision to publish, or preparation of the manuscript.

### Grant Disclosures
The following grant information was disclosed by the authors:
National Natural Science Foundation of China: 31601803.
Fundamental Research Funds for Central Non-profit Scientific Institution: 1610132020032.
China Agriculture Research System: CARS20.

### Competing Interests
The authors declare that they have no competing interests.

### Author Contributions
- Zhihao Hou conceived and designed the experiments, performed the experiments, analyzed the data, prepared figures and/or tables, authored or reviewed drafts of the paper, and approved the final draft.

- Qiang Chen performed the experiments, prepared figures and/or tables, and approved the final draft.
- Mengran Zhao analyzed the data, prepared figures and/or tables, and approved the final draft.
- Chenyang Huang conceived and designed the experiments, authored or reviewed drafts of the paper, and approved the final draft.
- Xiangli Wu conceived and designed the experiments, authored or reviewed drafts of the paper, and approved the final draft.

## DNA Deposition

The following information was supplied regarding the deposition of DNA sequences:

The coding sequences of PoZCP genes are available at GenBank: MN652925 to MN652990.

## Data Availability

Raw data are available in Tables S1–S5 and Data S1–S2.

## Supplemental Information

Supplemental information for this article can be found online at http://dx.doi.org/10.7717/peerj.9336#supplemental-information.

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
