# Peer review of "Genome-wide characterization of the Zn(II)2Cys6 zinc cluster-encoding gene family in Pleurotus ostreatus and expression analyses of this family during developmental stages and under heat stress"

_PeerJ, doi:10.7717/peerj.9336_

## Round 0.1 · original submission · Minor Revisions

I couldn't agree more with the reviewers. All major aspects, as well as sensitization to language issues, are named. I ask you to fulfill all the requirements of the reviewers.

Since we live more and more in the visual era it would be advantageous if you summarize your results/conclusions in some kind of an explanatory figure.

I keep my fingers crossed.

·

Basic reporting

Clearly reported and supported by references

Experimental design

The experimental design is good and authors have defined questions properly and performed rigorous investigations with proper experiement

Validity of the findings

overall good findings

Additional comments

Authors have extensively characterized 66 Zn(II)2Cys6 zinc cluster proteins (ZCPs) in Pleurotus ostreatus as PoZCPs and classified into 15 sub- types based on zinc cluster domain.
Authors also described the expression profiles of these PoZCP genes during different developmental stages and under heat stress as well. This work is good, however, I suggest following improvements
Authors must discuss:
1) What is known about these proteins in terms of number, gene structure and expression in other Agaricales.
2) How it is different from ascomycetes ?
3) What is the status of zinc cluster proteins in this species?


Minor changes
Authors are requested to properly edit the article for language issues, I have pointed out a few errors.
A. "Zinc cluster proteins were classified based on the amount of amino acid residues between the six cysteine residues of the typical zinc cluster domain"

to

"Zinc cluster proteins were classified based on the number of amino acid residues between the six cysteine residues of the typical zinc cluster domain"

B. "The domain region of zinc cluster sequences was characterized from the NCBI-CDD and Pfam results"
to
"The domain region of zinc cluster sequences was characterized by the NCBI-CDD and Pfam results"

C. "the poorly aligned regions were removed by alignment utilities trimAl (Capella-Gutiérrez et al., 2009). "
to
"the poorly aligned regions were removed by alignment utility trimAl (Capella-Gutiérrez et al., 2009). "

D. "this findings are similar to those in Aspergillus species and T. guangdongense, "
to
"these findings are similar to those in Aspergillus species and T. guangdongense, "

E. "For example, the proline located in the loop between the two substructures provides flexibility of the loop,"
to
"For example, the proline located in the loop between the two substructures provides the flexibility of the loop,"

·

Basic reporting

- The manuscript is clearly written overall; however, the text would benefit from minor revisions in language.
- The authors provide appropriate literature references, covering also some of the most recent literature relevant to their work.
- The manuscript is well structured and the figures and tables provided help the readers understand the main findings of this work. In particular, illustrations are clear and of high aesthetic value!
- The results reported adequately support the conclusions made by the authors.

Experimental design

- The authors present original research, annotating and further studying
the proteins predicted to be encoded by members of the Zn(II)2Cys6 zinc cluster gene family in Pleurotus ostreatus (PoZCPs).
- The authors characterise predicted protein coding genes in P. ostreatus using a semi-automated approach. Briefly, they automatically scan predicted polypeptide sequences with the respective PFAM hidden Markov Model (PF00172), and then proceed by removing redundant sequences and manually curating the remaining sequences before submitting to the NCBI CDD and PFAM for confirmation.
+ The "curation" and redundancy removal step is not clear why and how they were performed. This part of the Methods section needs to be more detailed.
+ To facilitate reproducibility, I would suggest that all candidate PoZCP sequences are reported. Those that have been discarded (redundancy/curation) could then be flagged appropriately, so interested readers could examine these intermediate findings as well.
- The authors further classify PoZCPs based on the lengths of the regions between the cystein residues in the zinc cluster domain. Even though this choice makes sense, it would be interesting to point to relevant literature or discuss the possible implications of these features in functional/structural properties of ZPCPs.
- Using MEME, the authors further identify conserved motifs within the reported sequences. However, apart of the presence of such motifs they do not mention whether any of those are related with other potentially known biologically important motifs (such as PFAM domains). In addition, it is unclear whether these motifs are specific to ZCPs or may appear in other protein classes as well within Pleurotus ostreatus, fungi in general or other species as well.
- RNA-seq was used to probe gene expression (under varying developmental or physiological conditions) and interesting patterns regarding different PoZCPs were identified. Even though experimental measurements for genes encoding PoZCPs are presented as supplementary material, I would expect that the raw RNA-seq data would be deposited in a public repository (e.g. GEO, Array Express) to facilitate re-use of this useful dataset.
- The authors further select a subset of 13 genes to be validated for gene expression by RT-qPCR. This is excellent, however the selection process is not clear: stating that they selected genes that "dramatically" changed their expression patterns (line 146) should be better substantiated.

Validity of the findings

- The authors report the fraction of all TFs falling under the ZCPs in P. ostreatus (line 157). However, they fail to mention how were the TFs annotated overall.
- The authors interestingly observe that the zinc cluster domains can be located anywhere along the protein sequences. However, qualifications such as N-terminal, C-terminal, and middle (lines 160-162) should be defined. It would be interesting to compare these findings with ZCPs in other studied species.
- The authors produce interesting sequence logos for the major types of ZCPs. I assume that "major" (line 179) here referes to types with >3 sequences. Please clarify.
- Importantly, sequence analysis predicts that the vast majority (62.12%) of PoZCPs is localized in the nucleus. However, the authors do not discuss the possible functions of PoZCPs that are localized in the cytoplasm or are secreted. Could these point to possible false positive characterizations of PoZCPs? In addition, please make clear that localization is inferred by prediction only (lines 187-189).
- The existence of motifs detected by MEME helps to see possible evolutionary relations of PoZCPs. Nevertheless, the essence of these motifs could be further explored (see my comment above).

Additional comments

This is a very interesting and well-performed work, combining dry and wet experiments. Some minor linguistic issues can be easily resolved by the authors, perhaps with assistance from a native English speaker.

---

## Round 0.2 · accepted · Accept

Congratulations.

I would ask you to make corrections identified by Reviewer 2 during the proof stage.

·

Basic reporting

Good

Experimental design

Overall good

Validity of the findings

Good.

Additional comments

After clear the manuscript from my side.

·

Basic reporting

The revised version of the manuscript is much improved and has addressed important issues raised during the initial round of review. Only a few issues remain to be resolved.

Experimental design

- Thank you for clarifying the curation and redundancy issue. It seems my lack of understanding was due to the use of the term "redundancy", which when used in the biomolecular sequence analysis context has a different meaning compared to colloquial English. When the instances of the matches PFAM motifs do not contain six cysteine residues it would be more appropriate to mention them as "putative false positives" rather than "redundant". Please make this correction to the supplementary material as well.

Validity of the findings

- Lines 183-188, and Table S3: Please indicate which tool(s) have been used to predict subcellular localization.

Additional comments

Thank you for your efforts in carefully revising the manuscript.